# Human *ARF* Specifically Inhibits Epimorphic Regeneration in the Zebrafish Heart

**DOI:** 10.3390/genes11060666

**Published:** 2020-06-18

**Authors:** Solomon Lee, Robert Hesse, Stanley Tamaki, Catharine Garland, Jason H. Pomerantz

**Affiliations:** 1Department of Surgery, Division of Plastic Surgery, Program in Craniofacial Biology, University of California, San Francisco, CA 94143, USA; solomon.lee@ucsf.edu; 2Department of Surgery and Orofacial Sciences, Program in Craniofacial Biology, Eli and Edythe Broad Center of Regeneration Medicine and Stem Cell Research, University of California, San Francisco, CA 94143, USA; robert.g.hesse@gmail.com (R.H.); stanley.tamaki@ucsf.edu (S.T.); 3Department of Surgery, Division of Plastic Surgery, University of Wisconsin School of Medicine and Public Health, Madison, WI 53726, USA; garland@surgery.wisc.edu; 4Department of Surgery and Orofacial Sciences, Division of Plastic Surgery, Program in Craniofacial Biology, University of California, San Francisco, CA 94143, USA; 5Edythe Broad Center of Regeneration Medicine and Stem Cell Research, University of California, San Francisco, CA 94143, USA

**Keywords:** *Alternative Reading Frame (ARF)*, cardiac regeneration, epimorphic regeneration

## Abstract

The Alternative Reading Frame (ARF) protein is a tumor suppressor encoded by the *Cyclin Dependent Kinase Inhibitor 2A* gene in mammals but not lower regenerative vertebrates, and has been previously implicated as a context-sensitive suppressor of regeneration in murine skeletal muscle and humanized ARF-expressing zebrafish fins. This study extends our investigation of the role of ARF in the regeneration of other solid tissues, including the zebrafish heart and the mammalian digit. Heart regeneration after cryoinjury was used to mimic massive myocardial infarction. *ARF* gene expression was upregulated during the cardiac regenerative process and slowed the rate of morphological recovery. *ARF* specifically impacts cardiomyocytes, neovascularization, and the endothelial-mesenchymal transition, while not affecting epicardial proliferation. This suggests that in the context of regeneration, *ARF* is specifically expressed in cells undergoing dedifferentiation. To investigate *ARF* as a suppressor of epimorphic regeneration in mammalian systems, we also tested whether the absence of *ARF* was permissive for murine digit regeneration, but found that *ARF* absence alone was insufficient to significantly alter digit restoration. These findings provide additional evidence that *ARF* suppresses epimorphic regeneration, but suggests that modulation of *ARF* alone is insufficient to permit regeneration.

## 1. Introduction

Humans do not undergo epimorphic regeneration after significant injury, which instead leads to collagen deposition, scar formation, and functional impairment. In contrast, zebrafish are able to fully regenerate complex structures such as the fin and heart even after massive injury from fin amputation [1], ventricular resection [2,3], genetic ablation [4], hypoxia-reoxygenation injury [5], and ventricular cryoinjury [6,7,8]. Zebrafish have thus served as excellent models to advance our knowledge of epimorphic regeneration and also elucidate limitations to human regeneration.

Tumor suppressor genes have increasing importance in cancer protection manifested most strongly in mammalian phyla, but may also adversely affect mammalian regeneration [9]. The human *Alternative Reading Frame* (*ARF*) gene product is an unusual tumor suppressor differentiated from others because it does not have any orthologs represented in highly-regenerative species [10,11], suggesting that the absence of *ARF* may be permissive for regeneration in those species. ARF is a protein encoded by *Cyclin dependent kinase inhibitor 2A* (*Cdkn2a*) [12] that responds to inappropriate retinoblastoma (Rb) pathway signaling [13] by sequestering Mouse double minute 2 (MDM2) to regulate Tumor protein 53 (p53) [14,15] and promoting cell cycle arrest or apoptosis to maintain the postmitotic state. Prior studies have implicated ARF as a regeneration suppressor in vitro by showing that blocking ARF in the context of compromise of the Rb pathway results in dedifferentiation and proliferation of mammalian muscle cells in culture [16]. In vivo in humanized *ARF* transgenic zebrafish, it was demonstrated that ARF is selectively activated during regenerative contexts to inhibit epimorphic fin regeneration after fin amputation [17]. Thus, experimental evidence supports the concept that ARF is a regeneration suppressor.

Although epimorphic regeneration exists in diverse tissues and a limited number of species, it is unknown how conserved epimorphic regeneration is mechanistically in different tissue and cell types. Likewise, it is unclear how generalizable are the mechanistic underpinnings of putative regeneration suppression by ARF, and by extension how dependent epimorphic regeneration is on the dedifferentiation of postmitotic cells. Zebrafish fin regeneration differs from cardiac regeneration at the histological and tissue levels. An amputated fin has a sharply demarcated wound edge that regenerates primarily using a blastema, a heterogenous pool of highly proliferative mesenchymal cells [2,18,19]. By comparison, the hallmarks of cardiac regeneration are remuscularization, angiogenesis, immunological balance, resolution of fibrosis, and electromechanical stability stemming from multiple tissue types rather than a singular blastema [20]. Cardiac cryoinjury in zebrafish results in immediate massive localized damage, and fibrotic tissue deposition similar to scarring after myocardial infarction in mammals [6,7]. All major cardiac tissue types including the epicardium, endocardium, and myocardium are then activated in an organ-wide response to the injury [21,22]. Epicardial cells throughout the heart initially induce developmental markers *T-Box transcription factor 18* (*tbx18*), *retinaldehyde dehydrogenase 2* (*raldh2*), *fibronectin 1* (*fn1*), and *Wilm’s tumor 1* (*wt1*), and undergo rapid proliferation to surround the exposed myocardium [21,23]. Activated epicardium near the injury site upregulates Sonic hedgehog (Shh), Notch, *platelet-derived growth factor receptor* β (*pdgfr*β)*, insulin-like growth factor* (*igf2b*), Bone morphogenetic protein (Bmp), Transforming growth factor β (TGFβ), and *fibroblast growth factor* (*fgfr*) to stimulate neighboring cardiomyocyte proliferation [3,21,23,24]. Epicardial cells also promote vasculature regeneration indirectly through paracrine mechanisms [25], although a subset undergoes epithelial-mesenchymal transition (EMT) and generates new vasculature directly [21]. Endocardial cells likewise upregulate *raldh2* as an initial response to injury [26], and subsequently contribute to cardiomyocyte regeneration [24]. Activated endocardial cells have also demonstrated an ability to induce atrial to ventricular myocardial transdifferentiation through the Notch pathway [27]. Cardiac myosin light chain 2 (Cmlc2^+^) cardiomyocyte proliferation is the primary source of myocardial regeneration [28]. The injury site regulates a variety of factors including retinoic acid [26], *tgfb1* [29], *pdgf* [30], Shh [31], *igf2b* [31,32] and Notch signaling [3,24] to promote cardiomyocyte proliferation. In response, existing cardiomyocytes near the injury undergo limited dedifferentiation with the disassembly of their sarcomeric structure and acquire immature phenotypes to facilitate mitosis before reconstituting the myocardium [28,33].

This study extends our investigation of the regeneration suppressive function of *ARF* in additional tissues, including the zebrafish heart and the mammalian digit. We demonstrate that *ARF* expression is upregulated during cardiac regeneration and slows the rate of morphological recovery, specifically impacting cardiomyocytes, neovascularization, and EMT, while not affecting epicardial proliferation. We also show that *ARF* deficiency alone is insufficient to permit significant digit regeneration.

## 2. Materials and Methods

### 2.1. Zebrafish and Mice

All experiments with zebrafish and mice were approved by the Institutional Animal Care and Use Committee of the University of California, San Francisco (ethical approval codes AN181101-01 and AN109021-03). Also, 3- to 6-month-old wild type (WT) AB or transgenic AB zebrafish were used for all cardiac regeneration experiments. The transgenic *hs*:ARF and *ARF*:ARF fish were from in-crossed transgenic lines used previously [17]. Five-month-old C57BL/6J male WT and *Arf* knockout (KO) mice were used for digit regeneration experiments.

### 2.2. Heat Shock Experiments

Heat shocks were delivered by housing fish in a water bath set to 37 °C with bidiurnal water exchanges. The water bath achieved 37 °C within 15 min, maintained that temperature for 1 h, and then passively cooled to fish room temperature (26–28 °C). An automatic digital timer (Intermatic, Spring Grove, IL, USA) was used to turn on and off the water bath. For heat shock experiments, an initial heat shock was delivered and then hearts were injured 3 h later. Heat shocks were subsequently delivered every 6 h for the duration of the experiment.

### 2.3. Cryoinjury

Cardiac cryoinjury and heart dissection were performed as previously described [7]. Sections with the most prominent cardiac injury from each sample were used. The injury site was approximated using Acid Fuchsin Orange G (AFOG) and troponin staining looking for fibrin, collagen, and disorganized myocardium. Myocardial recovery was calculated as the proportion of troponin infiltration into the demarcated injury site. Images were quantified using Adobe Photoshop CS6 software (Adobe, San Jose, CA, USA).

### 2.4. Mouse Anesthesia

Anesthesia was induced using 2–3% isoflurane with 0.8–1.0 L O_2_ flow into an induction chamber, watching for slow abdominal breathing and testing depth by paw pinch. Anesthesia was maintained using −2% isoflurane with 0.8–1.0 L O_2_ flow through a nose cone tubing. Isoflurane was decreased if there was agonal or mouth breathing. Animals were watched to full recovery after anesthesia and given analgesia if needed.

### 2.5. Digit Amputation

Mouse digit amputations were performed on a sterile field using a fresh 11-blade to make a clean cut. Mice were placed in the lateral position with forceps used to isolate the digit of interest. Digits were amputated either proximal or distal to the nail bed to emulate different regenerative processes. Pressure was applied after amputation to stop bleeding as needed. Amputated digits were collected at 10, 20, 30, and 60 days post-amputation (dpa) for analysis. Digits were amputated at the metacarpophalangeal joint (MCPJ) for harvest.

### 2.6. Histology and Immunofluorescence

Zebrafish heart section: Dissected, washed in phosphate buffered saline (PBS) containing 2 U/mL heparin and 0.1 M KCl, fixed in 4% paraformaldehyde (PFA) 1 h, equilibrated in 30% sucrose, frozen in optimal cutting temperature (OCT) blocks, sectioned at 10 µm, and stored at −80°C.

Mouse digit section: Dissected, fixed in 4% PFA overnight, washed in PBS 4–5 times, decalcified in 0.4M ethylenediaminetetraacetic acid (EDTA) for 2 weeks (except for Micro-CT specimens) at 4 °C, washed in PBS 4–5 times, frozen in 70% EtOH at −20 °C for storage, dehydrated, cleared, embedded in paraffin, sectioned at 5 µm.

AFOG staining: Rinsed in deionized (DI) water, fixed in Bouin’s fluid (Thermo, Waltham, MA, USA) at 60 °C 2 h, rinsed in DI water 30 min, submerged in Weigert’s iron hematoxylin (Thermo) solution 10 min, rinsed in DI water 5 min, submerged in 3% phosphotungstic-phosphomolybdic acid solution (Thermo) 5 min, rinsed in DI water 2 min, submerged in AFOG solution (5 g Methyl Blue boiled in 1000 mL DI water, 10 g Orange G, 15 g Acid Fuschin, pH adjusted to 1.09 by adding HCl) 5 min, rinsed in DI water 2 min, dehydrated in titrated EtOH baths and xylene, and mounted with toluene.

Trichrome staining: Sections were deparaffinized in xylene, EtOH washes, and H_2_O wash prior to staining. Mallory’s trichrome staining protocol was used.

Micro-CT: Mice were anesthetized and maintained with isoflurane for longitudinal X-ray and micro-CT studies of digit tip regeneration. X-ray images were obtained every 2–3 days using the scout view positioning function of a Scanco vivaCT-40 (Scanco, Brüttisellen, Switzerland) and full scans (55 kVp at 12.5 μm) were carried out on day 20.

Immunofluorescence: Washed in PBS with 0.1% Tween (PBST) 5 min, blocked with serum-free protein block (Dako) 1 h, primary antibodies in 5% goat serum antibody diluent at room temperature 4 h, PBST 5 min, secondary antibodies in antibody diluent 1 h, PBST 5 min, mounted with Vectashield mounting medium with or without DAPI (Vector Laboratories, Burlingame, Ca, USA). Myocyte enhancer factor 2 (Mef2)/Proliferating cell nuclear antigen (PCNA) staining was performed on sections as described [26]. Mef2^+^ and Mef2^+^/PCNA^+^ cells were counted manually. Primary antibodies are available in Table A1.

### 2.7. Quantitative Polymerase Chain Reaction (qPCR)

Cells were lysed with buffer RLT, and RNA was isolated using the RNeasy Plus Mini Kit (Qiagen, Hilden, Germany) according to the manufacturer’s protocol. cDNA was produced from total RNA using the SuperScript III First Stand Synthesis System (Life Technologies, Carlsbad, CA, USA) per the manufacturer’s protocol. Thermocycling and quantification were performed using the Mastercycler RealPlex 2 (Eppendorf, Hamburg, Germany). qPCR assays were performed on 10 ng of cDNA using 1.2 µL of each primer (10 pmol/mL) and iTaq Universal SYBR Green Supermix (Bio-Rad, Hercules, CA, USA) in a 12 µL total reaction volume. The PCR was performed for 40 cycles with annealing temperatures of 55–60 °C and elongation times of 20 sec. The relative expression of individual genes compared to control groups was calculated by the delta delta cycle threshold (ΔΔ-Ct) method with ribosomal protein S13 (RPS13) as the housekeeping gene. PCR primer sequences are available in Table A2.

### 2.8. Statistical Analysis

Data are presented as mean ± standard error. Statistical analyses were performed by using Stata Statistical Software, SE 12 (StataCorp, College Station, TX, USA). Statistical differences were analyzed by using the unpaired t-test, ANOVA, and Pearson’s correlation coefficient. A *p* < 0.05 was set as the threshold for statistical significance.

## 3. Results

### 3.1. Induced Expression of ARF Suppresses Cardiac Regeneration

To investigate whether human ARF assumes canonical functions in cardiac cells in vivo, we used fish in which ARF is expressed under the control of the heat shock protein 70 inducible promoter (Tg(*hsp70l*:ARF) or *hs*:ARF) [17]. In the fin of *hs*:ARF fish, a heat shock regimen of 1 h at 37 °C drives pervasive expression of ARF in cells, with peak ARF expression 3 h after heat shock [17]. qPCR confirmed that *ARF* mRNA is also expressed in the hearts of fish that underwent the same heat shock regimen (Figure 1a).

To evaluate the phenotypic impact of induced ARF expression on cardiac regeneration, the hearts of *hs*:ARF fish and non-transgenic WT clutchmates were cryoinjured 3 h after heat shock and were subsequently heat-shocked every 6 h to maintain ARF expression. *hs*:ARF and WT fish both tolerated heat shock without gross illness or mortality. Hearts were analyzed 15 days post injury (dpi). ARF expression was associated with significantly less myocardial recovery within the injury site compared to WT controls that underwent heat shock (Figure 1b). AFOG staining was used to visualize the injury site and the involved tissue types. With this technique, by 15 dpi in WT fish, the central part of the infarct area was replaced by a loose collagen network marked in blue, and a red fibrin layer formed in the inner margin of the wound was in the process of being replaced by myocytes at the edges, as previously described [6]. Compared to WT hearts that showed the expected regeneration phenotype, the *hs*:ARF hearts had a bulging disorganized collagen network, a less organized fibrin layer, and also less muscle tissue infiltration of the injury site (Figure 1b). The degree of myocardial recovery was better visualized by troponin immunostaining which showed significantly fewer troponin expressing myocytes in the *hs*:ARF injury site (Figure 1b). When troponin presence was quantified as a percentage of the injury site, WT recovery measured 57.8 ± 5.1% compared with *hs*:ARF recovery, which measured 29.7 ± 1.3%, a 48.7% reduction (*p* < 0.01) (Figure 1c).

### 3.2. ARF Expressed under Control of the Endogenous Human ARF Promoter Suppresses Cardiac Regeneration

To understand whether signals present in the regenerating heart would be detected by *ARF*, we evaluated the impact of ARF expression when under control of the native human *ARF* promoter using Tg(*ARF*:ARF), or *ARF*:ARF fish. The hearts of cryoinjured *ARF*:ARF and WT clutch mates were collected at 0, 1, 4, 7, 11, 15, 30, and 45 dpi. The regeneration of *ARF*:ARF hearts was significantly delayed compared to WT fish (Figure 2). AFOG staining was used to qualify the progression of cardiac regeneration over time. WT hearts followed expected patterns of myocardial regeneration [6] with the formation of a loose collagen network and thin fibrin layer replacing necrotic myocardium by 7 dpi, and progressive myocardial replacement of damaged tissue inward from the healthy edges with complete scar tissue replacement by myocardium at 30 dpi (Figure 2a). By comparison, *ARF*:ARF hearts were characterized by a much more disorganized fibrin deposit on 7 dpi, and from that point forward, attained consistently less myocardial replacement of the injury site (Figure 2a). Even by 30 dpi when most WT hearts had fully recovered, in *ARF*:ARF fish, fibrin persisted in the injury site, which is typically cleared by 21 dpi (Figure 2a) [6].

When quantified by proportion of troponin within the injury site, WT vs. *ARF*:ARF cardiomyocyte recovery measured 4.9 ± 1.8% vs. 4.8 ± 1.3%, 20.4 ± 2.8% vs. 16.2 ± 4.6%, 52.9 ± 3.9% vs. 31.0 ± 3.0%, 77.3 ± 6.3% vs. 49.4 ± 7.2%, and 95.0 ± 1.3% vs. 72.0 ± 4.3% on 1, 4, 7, 15, and 30 dpi, respectively. ANOVA testing showed a significant difference between WT and *ARF*:ARF myocardial recovery (*p* < 0.01). Stratifying by dpi, deficiencies in recovery of *ARF*:ARF hearts were 2.3% (*p* = 0.96), 20.4% (*p* = 0.47), 41.3% (*p* < 0.01), 36.1% (*p* = 0.05), and 24.3% (*p* < 0.01) at 1, 4, 7, 15, and 30 dpi, respectively (Figure 2b). In contrast to the fin, regeneration did methodically progress in *ARF*:ARF hearts despite the suppressed rate. This differed from the amputated fins of some *ARF*:ARF fish which never fully regenerated regardless of time [17]. All *ARF*:ARF hearts were fully restored by 45 dpi compared to WT hearts which had regenerated by 30 dpi.

### 3.3. ARF Suppresses Cardiomyocyte Proliferation, Vascular Formation, and EMT

Cardiomyocyte proliferation during heart regeneration was quantified by immunostaining *ARF*:ARF and WT hearts at 11 dpi with a cardiomyocyte nuclear marker (Mef2) and cell proliferation marker (PCNA) (Figure 3a). Cardiomyocyte proliferation index was calculated as a proportion of Mef2^+^/PCNA^+^ cells among all Mef2^+^ cells within the injury site. The cardiomyocyte proliferation index was reduced by 46.6% (*p* = 0.01) in *ARF*:ARF hearts compared to WT (Figure 3b).

To further elucidate the cell types and processes involved in delayed cardiac regeneration in *ARF*:ARF fish, RNA of *ARF*:ARF and WT hearts were collected at 11 dpi to measure the expression of several tissue-specific regeneration associated genes. *ARF* mRNA expression in *ARF*:ARF cryoinjured hearts was confirmed by qPCR (Figure 4a). Low levels of *ARF* mRNA were detected in uninjured hearts, which could indicate a low-level leakiness of the promoter, or elevated free E2F. A ligand-induced in myocardial cycling *fibroblast growth factor 17b* (*fgf17b*), a receptor-induced in epicardial cycling *fibroblast growth factor receptor 2c* (*fgfr2c*), and an endothelial growth factor expressed during vascular cycling *vascular endothelial growth factor Aa* (*vegfaa*) were measured [21,34]. *fgf17b* and *vegfaa* were reduced by 42% (*p* < 0.01) and 43% (*p* < 0.01), respectively, in *ARF*: ARF hearts compared to WT, reflective of decreases in myocardial regeneration, and vascular regeneration, respectively. There was no significant difference in *fgfr2c* expression (*p* = 0.44) supporting that epicardial healing is unperturbed. *twist1b*, a marker for EMT [35], was reduced by 55% (*p* < 0.01) in *ARF*:ARF hearts, suggesting altered EMT (Figure 4b).

*ARF*, *fgf17b*, and *twist1b* expression were also followed over time in regenerating *ARF*:ARF and WT hearts (Figure 4c). In WT hearts in the absence of the *ARF* transgene, both *fgf17b* and *twist1b* rose steadily after cryoinjury before peaking on 11 dpi and tapering down to baseline levels by 30 dpi. The expression pattern mirrored the histological recovery (Figure 2a), with the peak expression interval occurring between 7 to¬ 15 dpi, before returning to baseline. In contrast, *ARF*:ARF hearts had much more variable expression of *fgf17b* and *twist1b* (Figure 4d). *fgf17b* was never significantly increased compared to the injured baseline. *twist1b* was only significantly increased at 11 dpi (*p* = 0.04), and otherwise did not change significantly compared to uninjured baseline. *ARF* expression was tracked over the same period in *ARF*:ARF fish. *ARF* expression was significantly elevated between 7–30 dpi (*p* < 0.04) compared to baseline, a consistent increase of *ARF* expression throughout the regeneration process before tapering off at 45 dpi when even *ARF*:ARF hearts had nearly fully recovered. Changes in *ARF* expression between time points roughly correlated with inverse changes in *fgf17b* (r = −0.32, *p* = 0.13) and *twist1b* (r = −0.60, *p* < 0.01) expression (Figure 4e).

### 3.4. ARF Deficiency Is Insufficient to Permit Digit Regeneration in Mice

Since our data collectively suggest that *ARF* will impede naturally occurring epimorphic regeneration, we asked the converse question of whether the absence of *ARF* alone would be permissive. To test this, we examined murine digit regeneration in *Arf* KO mice. As an initial pilot, *Arf* KO mice matched with WT controls underwent amputation at a distal or proximal location on the distal phalanx. Distal amputation was distal to the nail bed and tested whether *Arf* deficiency affected normal regrowth when the proximal nail bed stem cells remained present. Proximal amputation was proximal to the nail bed to test whether *Arf* deficiency could permit epimorphic regeneration.

For amputations proximal to the nail bed, gross morphologic assessment and histology showed that while healing occurred with scar, neither WT nor *Arf* deficient digits underwent epimorphic regeneration of the digit or nail (Appendix A). For distal digit amputations where the nail bed stem cells were intact, there was also no significant difference between WT and *Arf* deficient digits in the extent or rate of regeneration (*p* = 0.56). Micro-computed tomography (micro-CT) likewise did not detect substantial differences in bone regeneration. *Arf* deficient digits did contain intramedullary sclerosis and disorganized periosteal osseous proliferation near the amputation margin compared to WT mice, but no mature ossification or cortication was noted by histology or micro-CT (Appendix A).

## 4. Discussion

In this extension of our initial findings that mammalian ARF is a suppressor of fin regeneration, we show this property of ARF to be similarly active in the context of heart regeneration. Our findings imply that similarities exist among epimorphic regeneration processes that produce common cues able to be specifically sensed by *ARF*, inducing ARF expression and inhibition of regeneration. Despite differences in cell and tissue types and in mechanical processes of regeneration, there is a commonality of post-mitotic reentry into a proliferative state that is likely to be the common aspect of epimorphic regeneration, which in mammalian systems is restricted to the context of tumorigenesis. The findings imply similarities between regeneration and tumorigenesis and point to possible incompatibilities of mammalian tumor suppressor mechanisms and epimorphic regeneration.

ARF expression specifically reduced cardiomyocyte proliferation by nearly 50% when expressed under the control of the *hs* promoter or the endogenous *ARF* promoter. As occurs after myocardial infarction, both WT and *ARF*:ARF hearts initially responded to cryoinjury with collagen formation and fibrin deposition over the injury site. In this acute phase of recovery during the first few days after injury, there was no significant difference between WT and *ARF*:ARF fish. The observation that differences became apparent between 7–30 dpi further shows that ARF does not affect the acute initial wound healing response, similar to the case of wound healing in the fin [17]. Rather, ARF selectively suppresses regeneration responses including cardiomyocyte proliferation. This is further demonstrated by the decreased cardiomyocyte proliferation index and *fgf17b* expression, suggesting that suppression of cardiomyocyte cycling is a primary, though not necessarily exclusive mechanism by which ARF inhibits heart regeneration.

The selective suppression of cardiomyocytes is particularly highlighted by the observation that *ARF* expression appears to have no impact on epicardium as reflected by *fgfr2c* expression. A difference between cardiomyocyte and epicardial proliferation in cardiac regeneration after injury relates to the fact that cardiomyocytes undergo limited dedifferentiation with the disassembly of their sarcomeric structure prior to proliferation [33], whereas epicardial cells rapidly proliferate without dedifferentiation to form a surrounding wound epidermis [21,22]. This suggests that *ARF* is specifically activated in cells undergoing dedifferentiation or at least post-mitotic reversal. The findings are consistent with our previous observations that ARF prevents dedifferentiation in muscle cells in culture [16] and is expressed in the proliferating blastema of an injured fin but not the surrounding epithelial tissue [17].

The triggers for *ARF* activation are further elucidated by the impact of *ARF* on EMT. EMT is a process during which polarized epithelial cells undergo biochemical changes to assume a mesenchymal phenotype [36]. In cardiac regeneration, EMT is important for generating new coronary vasculature via the epicardium [22]. Despite epicardial proliferation being unaffected in *ARF*:ARF fish, EMT is significantly reduced as reflected by *twist1b* expression. Since a significant portion of the new coronary vasculature is generated by EMT, it follows that *vegfaa* would also be expected to be reduced as observed in our study. This suggests that *ARF* is activated to suppress both dedifferentiating cells and cells undergoing EMT. By contrast, *fgfr2c* expressing epicardial cells, while providing critical signals for cardiac regeneration, do not undergo significant structural or cell-type change [23], and therefore we hypothesize they are thus not impacted by *ARF* during proliferation. Important considerations in future research should consider confirming the inferences of gene expression morphologically in situ via measures such as directly visualizing neovascularization or recording cardiac function tests to better characterize the functional impact of ARF.

Since dedifferentiation and metaplasia are common characteristics of tumorigenesis, our new findings further reinforce prior evidence that *ARF*, while a critical tumor suppressor, simultaneously opposes regeneration functions [9,17,37]. Despite being able to distinguish developmental from regenerative contexts [17], *ARF* is unable to distinguish epimorphic regeneration and tumorigenesis. However, an examination of regeneration in adult mouse digits, in which blastema formation does not occur, showed no significant change in the absence of *Arf*. This is not surprising since regeneration is a complex process unlikely to be enabled by a single gene. Furthermore, since our experiments in the fish fin and heart suggest that ARF interferes with proliferation in established blastemas, ARF would be expected to assume relevance in the mammalian context when regeneration progresses to the blastema stage.

## Figures and Tables

**Figure 1 genes-11-00666-f001:**
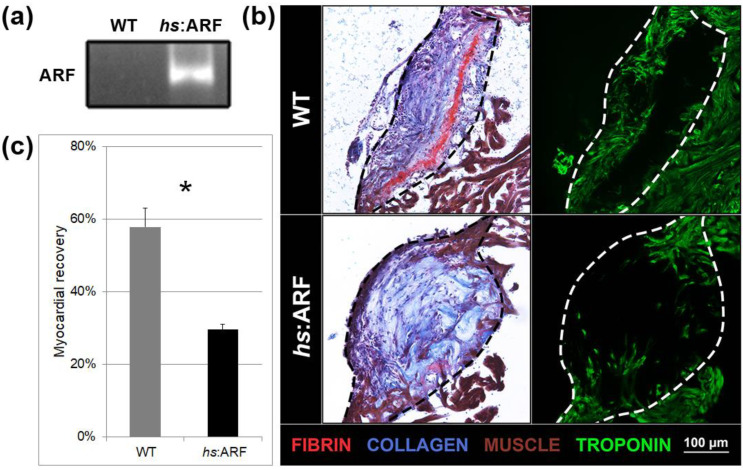
Induced Alternative Reading Frame (ARF) expression in *hs*:ARF fish suppresses cardiac regeneration. (**a**) Representative polymerase chain reaction (PCR) product of ARF expression from three experimental replicates of wild type (WT) and *hs*:ARF heart RNA on 15 dpi. The product run on gel electrophoresis shows ARF expression present in *hs*:ARF fish while absent in WT fish. (**b**) Acid Fuchsin Orange G (AFOG) and troponin staining of WT and *hs*:ARF heart cryosections on 15 days post-injury (dpi). The dotted lines demarcate the injury site. Red stains are for fibrin. Blue stains are for collagen. Brown stains are for muscle. Green stains are for troponin. (**c**) Myocardial recovery measured by troponin infiltration into the injury site. Infiltration quantified by imaging software shows decreased troponin in the injury site of *hs*:ARF fish compared to WT fish. N = 8 hearts. Results are shown as mean ± standard error. The * represents a statistically significant difference.

**Figure 2 genes-11-00666-f002:**
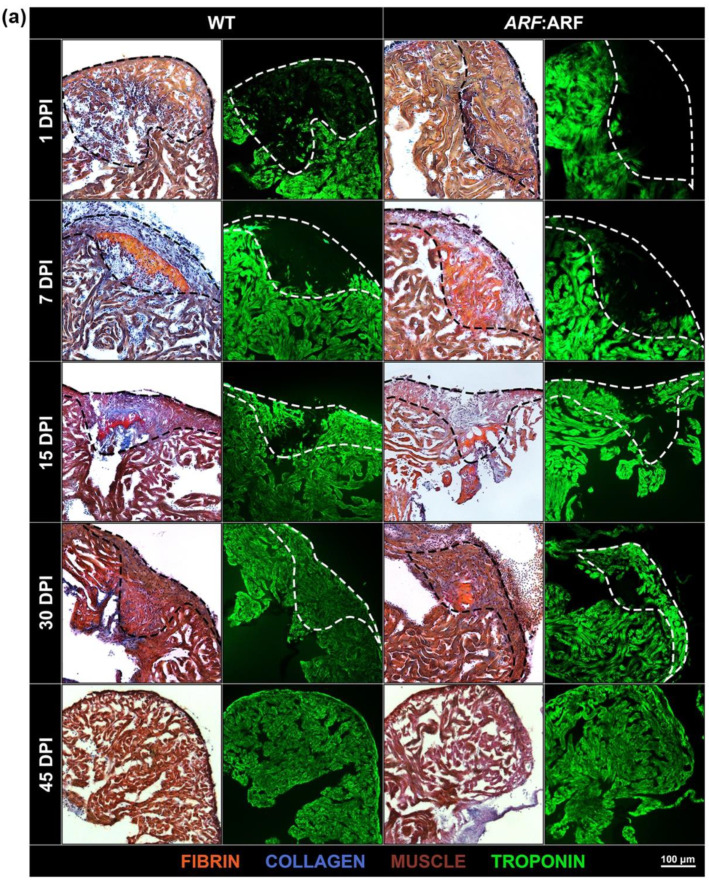
ARF expressed under control of the endogenous human *ARF* promoter in *ARF*:ARF fish suppresses cardiac regeneration over time. (**a**) AFOG and troponin staining of WT and *ARF*:ARF heart cryosections at 1, 7, 15, and 30 dpi. The dotted lines demarcate the injury site. Red stains are for fibrin. Blue stains are for collagen. Brown stains are for muscle. Green stains are for troponin. (**b**) Myocardial recovery measured by troponin infiltration into the injury site. Infiltration quantified by imaging software shows decreased troponin in the injury site of *ARF*:ARF fish compared to WT fish from 7 dpi onward. *N* = 49 hearts. The grey and black lines are logarithmic approximations of the WT and *ARF*:ARF heart recovery trends respectively. Results are shown as mean ± standard error. The * represents a statistically significant difference.

**Figure 3 genes-11-00666-f003:**
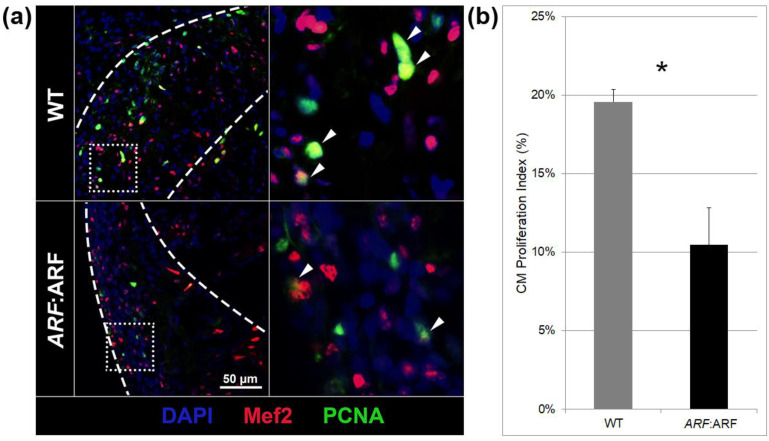
ARF expressed in *ARF*:ARF fish suppresses cardiomyocyte proliferation. (**a**) Myocyte enhancer factor 2 (Mef2), proliferating cell nuclear antigen (PCNA), and 4′,6-diamidino-2-phenylindole (DAPI) staining in WT and *ARF*:ARF fish at 11 dpi. Blue stains are for all nuclei. Red stains are for Mef2, a cardiomyocyte nuclear marker. Green stains are for PCNA, a cell cycling marker. (**b**) The cardiomyocyte (CM) proliferation index shows decreased cardiomyocyte cycling in *ARF*:ARF fish. The CM proliferation index was calculated by counting Mef2^+^/PCNA^+^ cells out of total Mef2^+^ cells in the injury zone. *N* = 8 hearts. Results are shown as mean ± standard error. The * represents a statistically significant difference.

**Figure 4 genes-11-00666-f004:**
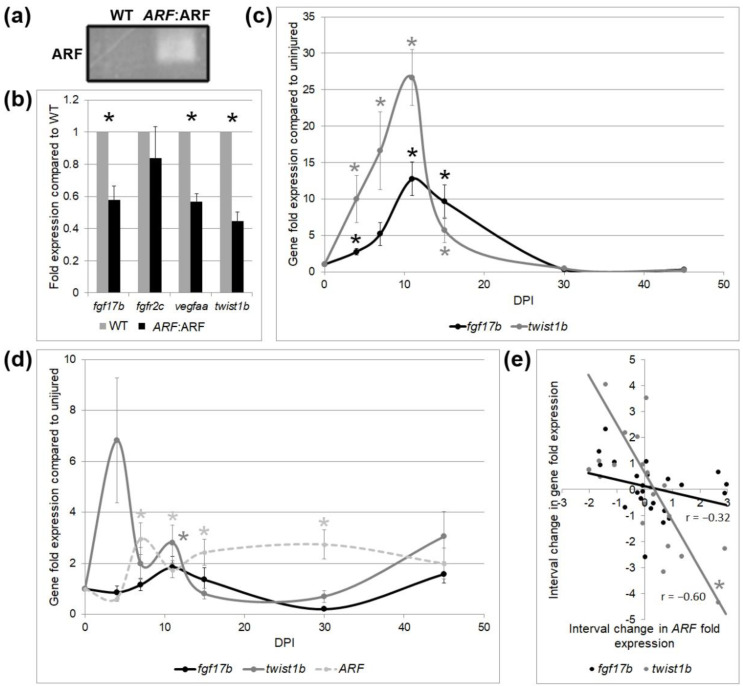
Tissue-specific gene expression by qPCR in WT and *ARF*:ARF fish over time. (**a**) Representative PCR product of *ARF* expression from three experimental replicates of WT and *ARF*:ARF heart RNA. The product run on gel electrophoresis shows *ARF* expression present in *ARF*:ARF fish while absent in WT fish at 11 dpi. (**b**) qPCR of *fibroblast growth factor 17b* (*fgf17b*), *fibroblast growth factor receptor 2c* (*fgfr2c*), *vascular endothelial growth factor Aa* (*vegfaa*), and *twist1b* mRNA expression for WT and *ARF*:ARF fish at 11 dpi. Results show a significant decrease in the expression of *fgf17b*, *vegfaa*, and *twist1b* in *ARF*:ARF fish. No significant difference occurred for *fgfr2c*. *N* = 8 hearts. (**c**) *fgf17b* and *twist1b* mRNA expression for WT fish over time compared to uninjured WT control. *fgf17b* and *twist1b* rise steadily after cryoinjury and peak by 11 dpi before tapering down to the uninjured baseline by 30 dpi. *N* = 18 hearts. (**d**) *fgf17b*, *twist1b*, and *ARF* mRNA expression for *ARF*:ARF fish over time compared to uninjured *ARF*:ARF control. *ARF* is significantly elevated above the uninjured baseline from 7–30 dpi. *fgf17b* is never elevated above the uninjured baseline. *twist1b* expression is only elevated above the uninjured baseline on 11 dpi. *N* = 24 hearts. (**e**) Correlation between interval change in *ARF* expression with changes in *fgf17b* and *twist1b* expression at any time point. *ARF* expression trended toward inverse correlation with *fgf17b* expression and was inversely correlated with *twist1b* expression. *N* = 24 hearts. Results are shown as mean ± standard error. The * represents a statistically significant difference.

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
