# Peer review of "Human ARF Specifically Inhibits Epimorphic Regeneration in the Zebrafish Heart"

_genes, 2020, doi:10.3390/genes11060666_

Round 1

Reviewer 1 Report

In this manuscript, Lee et al. demonstrate that the tumor suppressor, ARF, suppresses regeneration after cardiac injury. Using both zebrafish with a heat inducible ARFtransgene and humanized zebrafish that express ARFunder control of the endogenous human promoter, the authors show that ARFexpression suppresses cardiac regeneration after cryoinjury. They find that ARF does this by suppressing proliferation of cardiomyocytes and expression of genes involved in myocardial proliferation, neovascularization, and epithelial-to-mesenchymal transition (EMT). Additionally, they show that depletion of ARFin a murine model of digit regeneration was not permissive for regrowth, suggesting that other factors are involved in restricting regeneration in mammalian organisms.

Overall, the data is a nice extension of the authors previous work. However, the following issues should be addressed

  1. The authors need to ensure they are using the correct statistical analysis for each experiment. For example, the time courses in figure 2b cannot be compared using t-tests. An ANOVA should be used.
  2. It is intriguing that genes involved in EMT and vascularization are altered upon ARF exogenous expression. Is there evidence of reduced vascularization in the cardiac tissue of these fish?
  3. Text describing figure 4e needs to be added to the figure legend. For this panel, correlation coefficients and p-values should also be included to show a significant association.
  4. The authors suggest that ARF is only expressed in cardiomyocytes and this is responsible for the phenotypes being restricted to those cells. This should be demonstrated experimentally.

Author Response

Response to Reviewer 1 Comments

1. The authors need to ensure they are using the correct statistical analysis for each experiment. For example, the time courses in figure 2b cannot be compared using t-tests. An ANOVA should be used.

We thank the reviewer for this comment on the statistical analysis. Accordingly, we have performed a two-way ANOVA analysis on data for Figure 2b using ARF status and DPI as the independent variables and recovery as the dependent variable. Our analysis found ARF (p<0.01), DPI (p<0.01), and the interaction between ARF and DPI (p=0.02) all have statistically significant differences in heart recovery within the independent variable group. These revisions were made to the manuscript in lines 236-237. Methods were also updated in line 169 to include ANOVA analysis. However, as a two-way ANOVA test cannot differentiate which groups within variables are significantly different, we believe it is valuable to keep the t-tests comparing means between WT and ARF:ARF recovery when stratified by DPI. As these sub-analyses are comparing the means between two isolated groups and not the entire recovery curve, a t-test should remain statistically valid and provide nuance regarding when recovery is different for ARF:ARF fish.

2. It is intriguing that genes involved in EMT and vascularization are altered upon ARF exogenous expression. Is there evidence of reduced vascularization in the cardiac tissue of these fish?

Although we have tested and found reduced gene expression for important vascularization markers in cardiac tissues, we have not analyzed tissue morphology directly for reduced vascularization in ARF:ARF fish. We recognize that this is an important area to study in future experiments, and acknowledge that in the manuscript in lines 358-361.

3. Text describing figure 4e needs to be added to the figure legend. For this panel, correlation coefficients and p-values should also be included to show a significant association.

Thank you for the correction. Text describing Figure 4e has been added in the figure legend. Pearson’s correlation coefficient has also been calculated and correlation coefficients and p-values have been added to lines 298-299. The use of Pearson’s correlation coefficient has been added to line 169 in Methods. Notably, twist1b has a statistically significant negative correlation with ARF while fgf17b only trends toward negative correlation. We nevertheless included fgf17b as it supplements the statistically significant findings in Figure 4b.

4. The authors suggest that ARF is only expressed in cardiomyocytes and this is responsible for the phenotypes being restricted to those cells. This should be demonstrated experimentally.

Our data suggests that ARF is expressed in cardiomyocytes and not in the epicardium, but we did not intend to state that we demonstrate ARF is exclusive to cardiomyocytes. We have clarified this point by adding additional text to lines 336-337. Nevertheless, the reviewer raises a good point that if cardiomyocytes express ARF while other cell types do not, we should demonstrate this experimentally. Unfortunately, despite multiple efforts, we were not able to adequately demonstrate ARF expression in situ due to immunostaining difficulties in the heart.

Reviewer 2 Report

The paper entitled “Human ARF specifically inhibits epimorphic regeneration in diverse tissues but absence of ARF is not permissive in the absence of a blastema” that you kindly submitted for publication in the Journal “genes” now been considered. 

In the manuscript, the authors have identified of the involvement of the alternative reading frame (ARF) in regeneration of solid tissues like the zebrafish heart and the mouse digit by genetic modification of ARF expression at animal level.  One was conducted by getting more expression of ARF under cryoinjury followed by myocardial recovery in zebrafish heart upto 45 days.  The other was performed by mice digit amputation followed by collecting regenerating tissues upto 60 days in wild type and ARF KO mice.  The role of ARF overexpression in zebrafish heart was clear and solid to make a conclusion while the results from mice digit remains to be studied more in my opinion.  Overall, the study was interesting and new in terms of tissue regeneration and dedifferentiation

Specific comments:

  1. Overall, most studies in zebrafish heart recovery was about staining and expression of related genes. After figure 4, add functional activity of heart from WT and ARF/ARF zebrafish might be more conclusive and recommendable.
  2. The effect of ARF in mouse digit regeneration is not contributing to the conclusion enough and separate the part for another manuscript with further studies

Author Response

Response to Reviewer 2 Comments

1. Overall, most studies in zebrafish heart recovery was about staining and expression of related genes. After figure 4, add functional activity of heart from WT and ARF/ARF zebrafish might be more conclusive and recommendable.

We recognize the importance of assessing functional activity of the heart after regeneration as a useful marker for recovery to supplement our histologic studies. However, we were unable to assess cardiac function in detail because of a lack of readily available methods to assess active zebrafish myocardial function. We are able to say that ARF:ARF fish remained viable after cardiac injury until the end of the experiment. We articulate this point and its importance in future studies in the manuscript in lines 358-361.

The effect of ARF in mouse digit regeneration is not contributing to the conclusion enough and separate the part for another manuscript with further studies

We thank the reviewer for this comment. We recognize that the data for the mouse digit regeneration is preliminary and begins a new thread of research beyond the scope of this paper. In accordance with the reviewer’s suggestion, we have removed some content related to mouse digit regeneration from the main manuscript, changed the manuscript title, and revised the discussion to reflect those changes. The primary mouse figure has been added as supplementary material, as we believe it is still valuable in addressing a common and important question that is frequently asked in response to our zebrafish data.

Round 2

Reviewer 2 Report

 Although the functional aspects of heart impair were not provided in regards of ARF amplification in zebrafish model, they explained the reasons why they could not add the data.

I'm satisfied with revised manuscript.